# Preparation and Physicochemical Stability of Liquid Oral Dosage Forms Free of Potentially Harmful Excipient Designed for Pediatric Patients

**DOI:** 10.3390/pharmaceutics11040190

**Published:** 2019-04-18

**Authors:** Guillaume Binson, Karine Beuzit, Virginie Migeot, Léa Marco, Barbara Troussier, Nicolas Venisse, Antoine Dupuis

**Affiliations:** 1Health-Endocrine Disruptors-EXposome (HEDEX), CIC Inserm 1402, University Hospital of Poitiers, Poitiers 86021, France; guillaume.binson@chu-poitiers.fr (G.B.); virginie.migeot@univ-poitiers.fr (V.M.); n.venisse@chu-poitiers.fr (N.V.); 2Department of Pharmacy, University Hospital of Poitiers, Poitiers 86021, France; k.beuzit@chu-poitiers.fr (K.B.); lea.marco@hotmail.fr (L.M.); b.troussier@gmail.com (B.T.); 3School of Medicine and Pharmacy, University of Poitiers, Poitiers 86073, France; 4Department of Pharmacokinetics, University Hospital of Poitiers, Poitiers 86021, France

**Keywords:** liquid oral dosage form, physicochemical stability, potentially harmful excipient, pediatric patients, compounded preparation

## Abstract

Dexamethasone, hydrochlorothiazide, spironolactone, and phenytoin are commonly used in neonates, but no age-appropriate formulation containing these active pharmaceutical ingredients (APIs) is commercially available. Thus, pharmaceutical compounding of the liquid oral dosage form is required to enable newborn administration. A problem common to the compounded preparations described in the literature is that they include potentially harmful excipients (PHEs). Therefore, the aim of this study was to evaluate the feasibility of compounding oral liquid dosage forms free of PHE, containing dexamethasone, hydrochlorothiazide, phenytoin, or spironolactone and to assess their physicochemical stability. Due to the poor water solubility of the targeted APIs, oral suspensions were compounded using Syrspend® SF-PH4 Dry, a suspending vehicle free of PHE. Four HPLC coupled to UV spectrometry (HPLC-UV) stability-indicating methods were developed and validated according to international guidelines to assay the strength of the targeted APIs. Whatever storage condition was used (5 ± 3 °C or 22 ± 4 °C), no significant degradation of API occurred in compounded oral suspensions. Overall, the results attest to the physical and chemical stability of the four oral liquid dosage forms over 60 days under regular storage temperatures. Finally, the use of the proposed oral suspensions provides a reliable solution to reduce the exposure of children to potentially harmful excipients.

## 1. Introduction

Oral medicine administration to the pediatric population still remains a challenge, particularly due to a frequent lack of age-appropriate formulations, which raises a concern about dose accuracy [1]. Moreover, several factors specific to the pediatric population, such as the ability to swallow, palatability issues, etc., may hamper the administration of oral medication [2]. Indeed, most commercially available medicines are designed for adults and do not provide ease of use for children [3]. Beyond the efforts of health authorities to promote the development of pediatric medicines, many medicinal products are authorized only for adults and are not currently available in formulations suitable for administration to the pediatric population [4]. Consequently, caregivers or parents frequently modify medicines used off-label before administration in children, leading to dose error risk or inaccurate dosing, as well as stability and/or bioavailability issues pertaining to the drug [5]. In order to avoid manipulation of the dosage form, pharmaceutical compounding preparation may be proposed. Among the different formulations of compounded preparations available, liquid oral dosage forms represent an interesting alternative in children, providing increased dose flexibility and ease of swallowing compared to solid oral forms [4]. However, liquid dosage forms may require ingredients that are not child-friendly (suspending vehicles, preservatives, etc.).

Dexamethasone, hydrochlorothiazide, phenytoin, and spironolactone are four Active Pharmaceutical Ingredients (APIs) commonly used in neonates [6,7,8]. However, in many countries, formulations suitable for oral administration of these APIs to children are not currently commercially available. According to the literature, several methods to compound oral liquid preparations containing these APIs have been described [9,10,11,12,13,14,15]. These methods have used suspending vehicles formulated using suspending agents (sodium carboxymethylcellulose, xanthan gum), dispersant/humectants (propylene glycol, dimethicone), sweetening agents (sodium saccharin, sucrose), preservatives (methylparaben, propylparaben, sodium benzoate), and water as a solvent. A problem common to all these formulations is that they include potentially harmful excipients (PHEs), such as parabens or sodium benzoate. Exposure of neonates to PHEs is a safety concern since they are associated with toxicity issues, allergic reactions, and intolerance [16,17]. Indeed, sodium benzoate may induce severe hyperbilirubinemia due to its capacity to displace bilirubin from albumin [18]. Consequently, sodium benzoate is contraindicated in newborns less than eight weeks old. On the other hand, parabens are suspected of being endocrine disruptors [19]. Developmental exposure to endocrine-disrupting chemicals, during fetal or early life, is associated with a wide range of disorders and could increase the risk of disease later in life [20,21]. In this context, the European Medicine Agency (EMA) recommends avoidance of their use wherever possible, particularly in pediatric formulations [22]. Development of such formulations requires, as for any medicine, physicochemical stability studies, particularly, since PHE may have anti-oxidant properties, such as parabens or benzoic acid [23].

In light of this, the objective of this study was to evaluate the feasibility of compounding oral liquid dosage forms free of potentially harmful excipients, containing dexamethasone, hydrochlorothiazide, phenytoin, or spironolactone as APIs and to assess their physicochemical stability.

## 2. Materials and Methods

### 2.1. Chemicals and Reagents

Syrspend® SF PH4 Dry (batch number: 15H13-B03-312986) and spironolactone (batch number: 17E22-B11-335448) were purchased from Fagron (Thiais, France), hydrochlorothiazide (batch number: 005851) and phenytoin (batch number: 006520) were purchased from Inresa (Bartenheim, France), and dexamethasone acetate (batch number: 15090168/A) was purchased from Cooper (France). Sterile water was purchased from Fresenius (Versylene®, Sèvres, France).

HPLC-grade methanol and acetonitrile were purchased from Carlo Erba (Val-de-Reuil, France), and ultrapure water was provided using a Direct-Q UV3 water purification system (Millipore®, Guyancourt, France).

### 2.2. Feasibility Study

Oral liquid dosage forms were compounded at a target concentration of 5 mg/mL for spironolactone, dexamethasone, and phenytoin and 2 mg/mL for hydrochlorothiazide. Since these four APIs are poorly to very poorly soluble in water (0.089, 0.722, 0.032, and 0.022 mg/mL, for dexamethasone, hydrochlorothiazide, phenytoin, and spironolactone, respectively [24]), use of a suspending vehicle was mandatory in order to prepare a liquid dosage form. Syrspend® SF PH4 Dry, a powder ensuring correct drug suspension and designed to prepare oral liquid dosage form, was chosen since this marketed product is formulated mostly with starch, considered as an inert excipient, and does not contain any PHE.

The oral suspensions were prepared in accordance with the French guidelines [25] according to the following standard operating procedure:
-First, the exact amount of API powder was weighed in order to obtain the targeted concentration.-API powder was added to 13 g of Syrspend® SF PH4 Dry and triturated in a mortar until homogeneity was achieved.-Subsequently, sterile water was gently added, while stirring continuously, until reaching a final volume of 200 mL.-Finally, the suspension was bottled in a 20 mL amber type I glass container.


The entire procedure was carried out in a clean area, in order to limit microbiological contamination.

In order to verify that the formulations were providing correct dispersion of API in the suspending vehicle, the strengths of API in the compounded preparations were assessed. In this way, the concentrations (*n* = 6) of three different batches were determined for each oral liquid dosage form, using high-performance liquid chromatography coupled to UV spectrometry (HPLC-UV) methods described below. One milliliter oral suspension samples (*n* = 3), collected after shaking in order to obtain uniform dispersion of the API, were diluted in methanol (1:10, *v*:*v*), centrifuged at 3500 *g* for 10 min, and supernatants were then diluted in methanol (1:10, *v*:*v*) before injecting onto the column. According to the United States Pharmacopoeia, dose content uniformity is verified if every compounded preparation contains no less than 90% and no more than 110% of the theoretical concentration [26].

### 2.3. Stability Study

In order to investigate the influence of temperature on the physical and chemical stability properties of the oral suspensions, three bottles from different batches of each API were stored throughout the study duration either at room temperature (22 ± 4 °C) or under controlled refrigeration (5 ± 3 °C) (temperature was monitored daily).

API concentrations were determined at days 0, 7, 14, 30, 42, and 60, using the HPLC-UV methods described below. Oral suspension samples, collected after shaking in order to obtain uniform dispersion of the API, were diluted in methanol (1:10, *v*:*v*), centrifuged at 3500 g for 10 min, and supernatants were then diluted in methanol (1:10, *v*:*v*) before injecting onto the column.

According to the US Pharmacopoeia, compounded preparations are considered to be stable if the API concentration remains within 90–110% of the initial value (day 0) [27].

Osmolality and pH were also determined during the stability study, using a vapor pressure osmometer (VAPRO®, Wescor, Fontenay, France) and a pH-meter (EcoScan®, Eutech Instrument, Fontenay, France), respectively. In addition, physical appearance was investigated by visual inspection performed in a transparent glass vial, in order to check the initial color and opalescent aspect of the suspension.

### 2.4. Analytical Method Development and Validation

HPLC-UV analysis was performed using a chromatographic system, including a binary gradient solvent delivery pump (Hitachi L-2130®, VWR, Fontenay-sous-Bois, France) and an autosampler (Hitachi L-2200®, VWR, Fontenay-sous-Bois, France), connected to a UV-visible detector (SPD-6A®, Shimadzu, Marne-la-Vallée, France). Chromatographic data were recorded and processed using EasyChrom® software integrator (VWR, Fontenay-sous-Bois, France). The system was operated at ambient temperature. Injection volume was settled at 25 µL. For each API, a specific HPLC-UV stability-indicating method was developed and fully validated in order to determine API concentrations. The chromatographic conditions used for each API are shown in Table 1. Validation of each analytical method was performed in accordance with ICH Q2(R1) international guidelines using the following criteria: linearity accuracy (precision and trueness) and specificity [27].

#### 2.4.1. Calibration Curve

To assess API concentration in the compounded preparations, a calibration method was performed for each API. Calibration range was determined according to API target concentrations. For dexamethasone, phenytoin, and spironolactone, initial 100 µg/mL API solutions were prepared and then diluted using methanol to obtain a five-point calibration curve (0, 12.5, 25, 50, and 100 µg/mL). For hydrochlorothiazide, an initial 100 µg/mL solution was prepared and then diluted using methanol to obtain a six-point calibration curve (0, 12, 14, 20, 24, and 28 µg/mL). Calibration curves were generated by linear least-squares regression of the peak area versus API concentrations profiles.

#### 2.4.2. Linearity and Matrix Effect

Five calibration curves were prepared on five different days. Linearity was assessed through the analysis of correlation coefficient (*r*^2^), y-intercept and slope of the linear regression line, and residual values (expressed as the percentage of the theoretical value).

Matrix effect was assessed comparing calibration curves prepared in methanol versus calibration curves obtained using Syrspend® SF PH4 Dry. Dilution was performed using a methanol solution containing 10% of the suspending vehicle in order to mimic the exact composition of samples obtained after dilution of the oral suspensions before HPLC-UV analysis. The y-intercept and slope of linear regression lines were compared using a Student t-test (α = 0.05) to assess matrix effect.

#### 2.4.3. Accuracy and Limit of Quantification

Accuracy was investigated by assessing the precision and trueness of the method. For each API, quality controls (QC) were prepared in methanol at a level of 50 µg/mL for dexamethasone, phenytoin, and spironolactone and 20 µg/mL for hydrochlorothiazide, according to diluted oral suspension concentrations. Precision was assessed through the determination of the relative standard deviation (RSD) of the mean concentration determined for each QC, on the same day for repeatability (*n* = 6) and over three days for intermediate precision (*n* = 18). Trueness was assessed by determination of the percent recovery of the expected concentrations of QC used during the precision study. The limits of quantification (LOQ) were settled as the lowest point of the corresponding calibration curves.

#### 2.4.4. Specificity and Stability-Indicating Performance

The stability-indicating capability of the method was evaluated by verifying that degradation products did not coelute with intact APIs. To ensure that no degradation products were coeluting with API, a photodiode array detector (Waters 2996®, Guyancourt, France) was used to obtain a three-dimensional chromatogram. Each API oral suspension was subjected to severe stress (heat, acidic and basic conditions, oxidation) in accordance with ICH Q1 international guidelines. Samples were then injected onto the column to compare chromatograms to those obtained with non-degraded suspension and to perform peak purity check test.

## 3. Results

### 3.1. Method Validation

A specific HPLC-UV method was developed and fully validated for each of the four APIs. Calibration curves provided adequate linearity over the studied ranges since correlation coefficients were greater or equal to 0.9992 and residual values were lower, less than or equal to 5.0% (Table 2). Furthermore, the mean slope and y-intercept values of the calibration curves obtained using Syrspend® SF PH4 Dry were not statistically different from those obtained using methanol (*p* were all greater or equal to 0.17), evidencing the lack of matrix effect using this suspending vehicle. In such a case, calibration curves may be performed using methanol dilution of API standard solutions, rendering HPLC-UV analysis easier and faster.

Regarding precision and trueness, acceptance criteria were met for the four analytical methods developed, since RSD of the mean concentrations were less than or equal to 4.4% and the percent recovery of QC concentrations was close to 100% of the expected value (Table 3). The LOQ in the suspensions was settled at 1.25 mg/mL for dexamethasone, phenytoin, and spironolactone and 1.20 mg/mL for hydrochlorothiazide. These results demonstrate that the analytical methods developed in this paper are accurate and allow reliable determination of dexamethasone, phenytoin, hydrochlorothiazide, or spironolactone content in oral liquid dosage forms compounded using Syrspend® SF PH4 Dry.

During the forced degradation study, several degradation products were formed (Figure 1, Figure 2, Figure 3 and Figure 4) but none of them interfered with the API peaks in terms of retention time (resolution was always greater than 1.5) and according to UV spectrum analysis and purity check test of the peaks (Table 4). Given that all methods were linear and accurate and that no degradation products eluted with the API peak, these methods are stability-indicating according to international guidelines and can be used to determine targeted API content over time without interference by degradation products [27].

### 3.2. Feasibility and Stability Studies

Subsequent to the preparation, the mean concentrations of the compounded oral suspensions were equal to 4.70 ± 0.10 mg/mL, 2.0 ± 0.04 mg/mL, 4.91 ± 0.10 mg/mL, and 4.74 ± 0.06 mg/mL for dexamethasone, hydrochlorothiazide, phenytoin, and spironolactone, respectively. Thereby, these results demonstrate that drug loss during preparation was minimal.

Regarding chemical stability, irrespective of the type of the storage condition used, no significant degradation of dexamethasone, hydrochlorothiazide, phenytoin, or spironolactone occurred in the compounded oral suspensions (Figure 5). Indeed, during the period of testing, the proportion of the initial API concentration remaining was higher than 90%, meaning that oral suspensions were chemically stable up to 60 days under regular storage temperatures (at 5 ± 3 °C or at 22 ± 4 °C). In accordance with this result, no degradation products were observed over the assessed period. Moreover, for all samples, API concentration remained within 90 to 110% of the targeted strength, highlighting the fact that the dose content remained uniform over time for all the compounded suspensions.

According to the physical stability study, no color modification was observed, and no precipitates were retained over the storage period (Table 5). The pH of the oral suspensions remained stable over the testing period (less than 0.4 pH unit variation) (Figure 6), and no significant variation of osmolality occurred during the study (less than 100 mOsm/kg) (Table 5).

Overall, these results attest to the physical and chemical stability of the four oral liquid dosage forms over 60 days regardless of the type of storage conditions.

## 4. Discussion

An HPLC-UV method was developed for each targeted API and validated according to well-recognized international guidelines. These stability-indicating methods are suitable to investigate the physicochemical stability of dexamethasone, hydrochlorothiazide, phenytoin, or spironolactone in compounded preparations. Moreover, these methods, easy to implement and rapid to perform, may be routinely used for QC testing in order to perform a batch release of such compounded preparations.

To enable the appropriate volume of administration (preferably ≤1 mL for a newborn), according to the dose required in their therapeutic uses [6,8,28], we chose to prepare oral liquid dosage forms at a target concentration of 5 mg/mL for spironolactone, dexamethasone, and phenytoin and 2 mg/mL for hydrochlorothiazide. According to the literature, other authors have reported similar concentrations for oral liquid compounded preparations of the same APIs [10,13,15]. Some authors have proposed oral liquid dosage forms with lower strengths due to different therapeutic uses [12,14] or higher strengths since they were used in older children [9,11].

Dexamethasone, hydrochlorothiazide, phenytoin, and spironolactone are hydrophobic compounds displaying poor solubility in water, and they are unsuitable for the preparation of solutions at the desired concentrations. Consequently, we had to compound suspensions in order to provide oral liquid dosage forms. One major disadvantage of the suspension dosage form is the risk of the settling of API particles producing non-homogeneous preparation leading to inaccuracy in dosage measurement. Thus, suspensions must be shaken before use in order to obtain re-dispersion of API and to avoid dosage variability during oral administration. However, according to the feasibility study, dose content uniformity of the suspensions was evidenced since the deviation of the concentration of the API in the compounded preparations was never higher than 8% of expected API strength. Thereby, these results demonstrate that the oral formulations compounded using Syrspend® SF PH4 Dry as a suspending vehicle provide a reliable dosage of the four APIs. 

Moreover, the compounded liquid formulations display low osmolality (<100 mOsm/kg) compared to several other formulations (around 1600 mOsm/kg) [9,10,12], ensuring safe oral administration in neonates. Indeed, high osmolality, greater than 450 mOsm/kg, may lead to necrotizing enterocolitis in neonates, especially in premature newborns [29].

Regardless of the type of storage conditions, the proportion of the initial concentration of dexamethasone, hydrochlorothiazide, phenytoin, or spironolactone remaining was within the limit set by the U.S. Pharmacopeia [26], meaning that the oral suspensions of each drug compounded using Syrspend® SF PH4 Dry were stable up to 60 days both at room temperature or under refrigeration. This result is similar to or greater than the stability period reported by others [9,10,11,12,13,14,15]. The physicochemical stability of the four oral liquid dosage forms was investigated for a period of 60 days since the microbiological stability of suspensions compounded using Syrspend® SF PH4 Dry has been demonstrated by the manufacturer over the same period [30]. However, since the formulations do not contain any preservative, it is preferable to recommend storage under controlled refrigeration, in order to limit microbiological growth.

Young children and *a fortiori* neonates are uniquely vulnerable to PHE due to organ immaturity, specific pharmacokinetics, and since excipients may cause problems in the developing organism [31,32]. Several studies have demonstrated that neonates receive a significant number of potentially harmful pharmaceutical excipients, and therefore it is important to explore substitution possibilities [33,34]. To our knowledge, the study presented in this paper is the first to assess the feasibility of compounding dexamethasone, phenytoin, hydrochlorothiazide, and spironolactone oral suspensions using excipients free of potentially harmful effects.

Vehicles used to compound oral liquid medications of many APIs reported in the literature, including dexamethasone, phenytoin, hydrochlorothiazide, and spironolactone, contain potentially harmful excipients, especially preservatives. Indeed, since the liquid formulations present a risk of microbiological instability, preservatives are added to liquid compounded preparations. Among preservatives, parabens are the most widely used in drug formulation since they are considered safe, cheap, inert, and with no odor or taste [35]. Nevertheless, parabens are known to bind to estrogen receptors providing endocrine-disrupting properties [36]. Moreover, early life exposure to endocrine disrupting chemicals has been pointed out as a major concern regarding child health [37,38]. Given these elements, the EMA recommends to limit or even to avoid the use of parabens in medicine formulations, especially when intended for use in the pediatric population [22]. Once again, to our knowledge, the present study is the first to demonstrate the physicochemical stability of oral liquid dosage forms containing the targeted API and compounded without preservatives, such as parabens. This result is of particular interest since the lack of preservatives may influence the chemical stability of a compound, especially due to their potential antioxidant activity [23].

In conclusion, the data provided in this study demonstrate that compounding oral liquid dosage forms free of potentially harmful excipient is workable, at least for the targeted active pharmaceutical ingredients: dexamethasone, hydrochlorothiazide, phenytoin, and spironolactone. Moreover, the physicochemical stability of these oral liquid formulations is in accordance with common clinical use in children, including neonates. Even though the microbiological stability of oral suspensions compounded using Syrspend® SF PH4 Dry has been assessed by the manufacturer, further experiments should be performed in order to assess the in-use microbiological stability of the preparations compounded with dexamethasone, hydrochlorothiazide, phenytoin, or spironolactone.

Finally, use of the proposed oral suspensions provides a reliable solution to reduce the exposure of children, especially newborns, to potentially harmful excipients, including potentially endocrine-disrupting excipients, such as parabens.

## Figures and Tables

**Figure 1 pharmaceutics-11-00190-f001:**
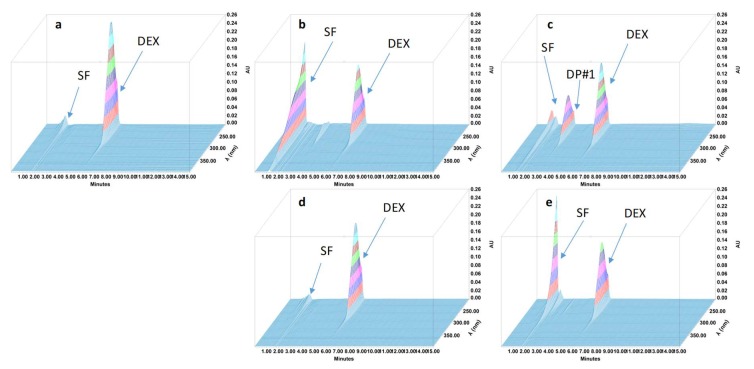
Three-dimensional chromatograms obtained for dexamethasone (DEX) oral suspensions after applying different stress conditions. (**a**) No stress; (**b**) HCl 0.5 M at 80 °C for 30 min; (**c**) NaOH 0.1 M at 80 °C for 10 min; (**d**) H_2_O_2_ 3% at 80 °C for 4 h; (**e**) 80 °C for 4 h. SF: solvent front; DP: degradation product.

**Figure 2 pharmaceutics-11-00190-f002:**
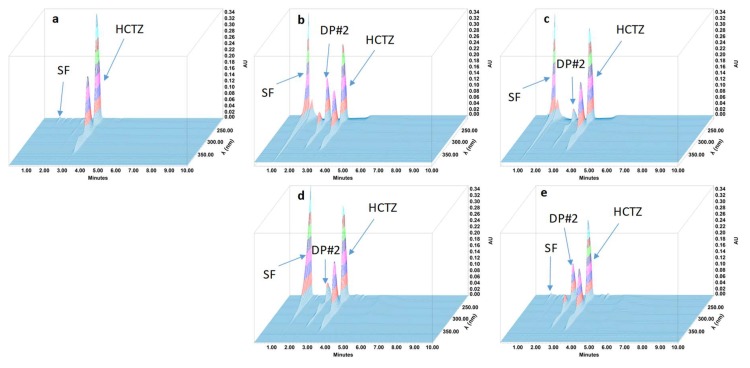
Three-dimensional chromatograms obtained for hydrochlorothiazide (HCTZ) oral suspensions after applying different stress conditions. (**a**) No stress; (**b**) HCl 1 M at 80 °C for 1 h; (**c**) NaOH 1 M at 80 °C for 1 h; (**d**) H_2_O_2_ 3% at 80 °C for 4 h; (**e**) 80 °C for 4 h. SF: solvent front; DP: degradation product.

**Figure 3 pharmaceutics-11-00190-f003:**
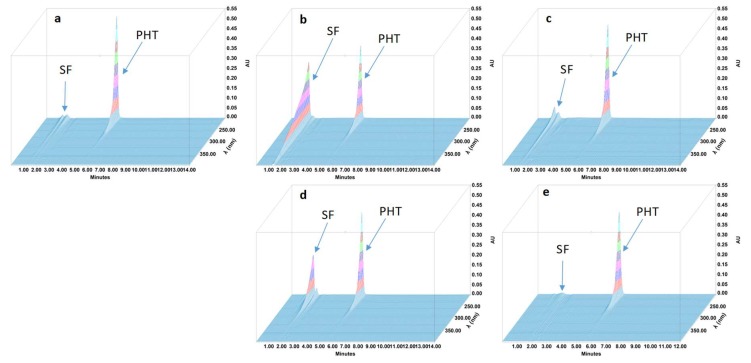
Three-dimensional chromatograms obtained for phenytoin (PHT) oral suspensions after applying different stress conditions. (**a**) No stress; (**b**) HCl 0.5 M at 80 °C for 4 h; (**c**) NaOH 1 M at 80 °C for 4 h; (**d**) H2O2 3% at 80 °C for 4 h; (**e**) 80 °C for 4 h. SF: solvent front.

**Figure 4 pharmaceutics-11-00190-f004:**
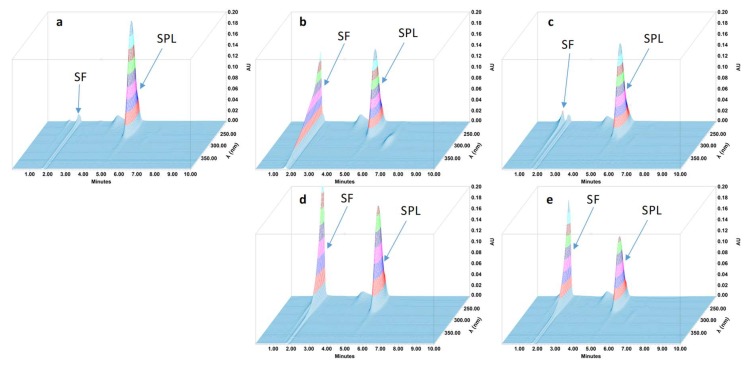
Three-dimensional chromatograms obtained for spironolactone (SPL) oral suspensions after applying different stress conditions. (**a**) No stress; (**b**) HCl 0.5 M at 80 °C for 1 h; (**c**) NaOH 0.1 M at 80 °C for 5 min; (**d**) H_2_O_2_ 3% at 80 °C 4 h; (**e**) 80 °C for 4 h. SF: solvent front.

**Figure 5 pharmaceutics-11-00190-f005:**
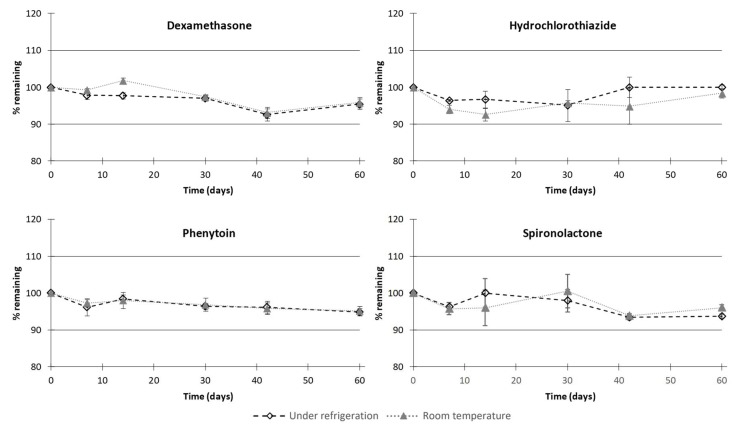
Chemical stability of dexamethasone, hydrochlorothiazide, phenytoin, and spironolactone in compounded oral suspensions in all storage conditions. Values are expressed as mean percentage remaining ± standard deviation.

**Figure 6 pharmaceutics-11-00190-f006:**
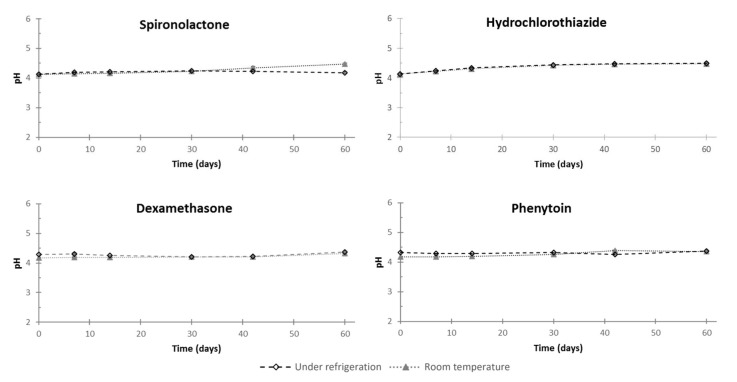
pH modifications of dexamethasone, hydrochlorothiazide, phenytoin, and spironolactone oral suspensions in all storage conditions. Values are expressed as mean ± standard deviation (standard deviation bar charts are within symbols of the mean).

**Table 1 pharmaceutics-11-00190-t001:** Characteristics of the different analytical methods developed to perform dexamethasone, hydrochlorothiazide, phenytoin, or spironolactone assay.

	Dexamethasone	Hydrochlorothiazide	Phenytoin	Spironolactone
**Mobile Phase**	Acetonitrile and water (50:50, *v*:*v*)	Methanol and water (20:80, *v*:*v*) pH adjusted to 4.5 using acetic acid	Acetonitrile and water (60:40, *v*:*v*)	Methanol and water (70:30, *v*:*v*)
**Flow rate (mL/min)**	1.0	1.5	1.0	1.0
**Column**	Purospher® STAR RP-18 endcapped (5 µm) 150 × 4.6 mm			
**λ (nm)**	238	224	238	238

**Table 2 pharmaceutics-11-00190-t002:** Performance parameters obtained during linearity assessment of the different analytical methods developed to perform dexamethasone, hydrochlorothiazide, phenytoin, or spironolactone assay.

	Dexamethasone	Hydrochlorothiazide	Phenytoin	Spironolactone
**Calibration range (mg/mL)**	1.25–10	12–28	1.25–10	1.25–10
***r*^2^**	0.9996	0.9996	0.9998	0.9992
**Maximum residual value**	5.0%	2.4%	3.5%	5.0%
**S/N**	438	138	349	164

S/N: signal to noise ratio at the lowest standard concentration.

**Table 3 pharmaceutics-11-00190-t003:** Performance parameters obtained during accuracy assessment of the different analytical methods developed to perform dexamethasone, hydrochlorothiazide, phenytoin, or spironolactone assay.

Colonne1	Spironolactone	Dexamethasone	Hydrochlorothiazide	Phenytoin
**Target concentration (mg/mL)**	5.0	5.0	2.0	5.0
**Mean concentration (mg/mL)**	*Repeatability*			
	4.9	5.0	2.0	5.0
**RSD (%)**	4.4	1.3	3.6	4.0
**CI 95% (%)**	[4.2 ; 4.7]	[1.3 ; 1.4]	[3.5 ; 3.6]	[1.9 ; 6.0]
**Mean concentration (mg/mL)**	*Intermediate precision*			
	5.0	4.9	2.0	5.0
**RSD (%)**	3.0	3.2	4.0	3.1
**CI 95% (%)**	[2.9 ; 3.1]	[3.1 ; 3.3]	[3.9 ; 4.0]	[2.3 ; 3.8]
**Recovery rate (%)**	*Trueness*			
	100.5	98.8	99.8	99.7
**95% CI (%)**	[99.0 ; 102.0]	[97.3 ; 100.4]	[97.8 ; 101.7]	[98.2 ; 101.2]

CI: confidence interval; RSD: relative standard deviation.

**Table 4 pharmaceutics-11-00190-t004:** Attributes related to three-dimensional chromatogram peaks obtained during a forced degradation study.

	Retention Time (min)	RRT	Tailing Factor	Resolution	Theoritical Plates
**Dexamethasone**	6.2	NA	1.03	NA	6797
*Degradation product #1*	3.2	0.52	1.12	12.74	6018
**Hydrochlorothiazide**	3.12	NA	1.16	NA	5399
*Degradation product #2*	2.25	0.72	1.18	6.06	5713
**Phénytoïne**	5.44	NA	1.17	NA	6240
	All degradation products were in solvent front				
**Spironolactone**	5.23	NA	1.16	NA	2547
	All degradation products were in solvent front				

NA: not applicable; RRT: relative retention time to the parent compound.

**Table 5 pharmaceutics-11-00190-t005:** Physical parameters assessed during the stability study of dexamethasone, hydrochlorothiazide, phenytoin, and spironolactone oral suspensions regardless of the type of storage conditions used (under refrigeration or at room temperature).

	Color		Precipitate	Osmolality (mOsm/kg)	
	Day 0	Day 60	Day 60	Day 0	Day 60
**Dexamethasone**	white opalescent	white opalescent	no	<LOQ	<LOQ
**Hydrochlorothiazide**	off-white opalescent	off-white opalescent	no	<LOQ	<LOQ
**Phenytoin**	white opalescent	white opalescent	no	<LOQ	<LOQ
**Spironolactone**	white opalescent	white opalescent	no	<LOQ	<LOQ

LOQ: limit of quantification (100 mOsm/kg).

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
