# Peer review of "Preparation and Physicochemical Stability of Liquid Oral Dosage Forms Free of Potentially Harmful Excipient Designed for Pediatric Patients"

_pharmaceutics, 2019, doi:10.3390/pharmaceutics11040190_

Round 1

Reviewer 1 Report

This research presents a valuable piece of work - have attached the document with some minor corrections and edits as well as questions.  

What I need to see included in the introduction and reflected on in the discussion is the stability of these selected APIs compounded with PHEs as these components  are unlikely to impact on the stability of the individual APIs. It is important that this body of work is included the introduction or to the extent that it is available. 

Author Response

Thank you very much for your valuable and constructive suggestions. According to reviewer’s suggestions we have included this point in the introduction (lines 65-67) as well as in the discussion (lines 367-369).

Moreover, all the comments provided in the pdf file have been taken in consideration.

Reviewer 2 Report

Dexamethasone, hydrochlorothiazide, phenytoin and spironolactone are commonly used in neonates but no oral liquid formulation is commercially available for children’s usage. Usually oral formulation compounding is associated with potential harmful excipients. In order to enhance the stability of liquid oral prescriptions, usually preservatives are added such as parabens, which is harmful for children. Therefore, in the present study, authors are investigating the feasibility of compounding these four mentioned active pharmaceutical Ingredients’ (API) oral formulation that is free of potential harmful excipients. In this study, authors developed and validated HPLC-UV methods for testing the stability of dexamethasone, hydrochlorothiazide, phenytoin or spironolactone. Results from these stability studies performed at room temperature and cold temperature indicated that oral formulations of these four APIs compounded using Syrspend® 255 SF PH4 Dry were stable up to 60 days. Herein, authors were successful in establishing the feasibility of the compounding oral liquid formulations of all four mentioned APIs without the use of any potential harmful excipients such as preservatives.  

Author Response

We sincerely appreciate the comments of the reviewers.

Reviewer 3 Report

This manuscript presented suspension formulations of four drugs that are devoid of harmful excipients for pediatric population. HPLC methods are also developed for each drug and validated for few parameters (not extensive). The formulation development and analytical methods are not presented with required scientific rigor. No chromatograms for samples containing degradants are supplied making it hard to comment on the validity or the quality of the methods presented. Further, the formulation development has not taken place methodically. Incorporation of an existing suspending agent with drug (a two component system) is not very novel. The reproducibility of such an approach may be problematic because an in-depth investigation of drug physical properties including solid form, particle size, etc. on the suspensions is not studied. Stability is also monitored for a limited time with assay parameters. It is unknown if any degradants are formed during stability study and whether there are acceptable limits for degradants. Overall, the manuscript is presented with minimal details and not suitable to publication in its current format.

The novelty, significance, comparison with existing formulations of these drugs, and previous attempts made by authors for pediatric population should be elaborated. The authors also seem not be very careful with formatting the paper. Some specific comments are show below.

1. Line 36 and 47. Sentences end with “…” This is not acceptable sentence formation.

2. Sedimentation rates and particle sizes are not measured and monitored during stability studies.

3. Requirement for cold storage is not very attractive.

4. Line 128 - incomplete reference and fix spacing!

5. 2.4.2. - This reviewer thinks that the best way to assess matrix effect would be to spike drug solution into P4 dry and then extract with methanol similar to samples!

6. What is the signal to noise ratio at the lowest concentration or so called LOD?

7. Lines 174/175 and 186. “Error!” message. Recheck.

8. Degradation study: Supply chromatograms! What is the resolution of main peak from degradants? Provide a table with chromatogram attributes supporting your validation including RRTs for major degradants, resolution, tailing factor, theoretical plates, etc.

Author Response

Thank you very much for your valuable and constructive suggestions. We agree with your opinion and we have carefully revised the manuscript according to the suggestions.

More details have been added to the revised manuscript regarding method validation (Tables 2), Figures 2,3,4,5,6), degradation products (lines 270-271), significance (367-369), and comparison with existing formulations (lines 52-55, lines 335-336, lines 342-343). The initial manuscript already contains several elements regarding novelty (lines 353-357, lines 365-367).

“Some specific comments are show below.”

1. Line 36 and 47. Sentences end with “…” This is not acceptable sentence formation.

The sentences has been corrected.

2. Sedimentation rates and particle sizes are not measured and monitored during stability studies.

The aim of the study was to assess physicochemical stability including uniformity in order to ensure that the formulations would provide safe and reliable administration of the targeted drug. Therefore, we did not characterized all physical parameters of the formulation. However, according to the results obtained during uniformity assessment and to visual inspection no significant modification was observed regarding sedimentation or precipitates formation.

3. Requirement for cold storage is not very attractive.

We agree that cold storage is not very attractive but this disadvantage is mandatory in order to avoid preservatives.

4. Line 128 - incomplete reference and fix spacing!

The bug has been fixed in the formatted manuscript.

5. 2.4.2. - This reviewer thinks that the best way to assess matrix effect would be to spike drug solution into P4 dry and then extract with methanol similar to samples!

Indeed, it would have been easier to do so, but the objective was to check that calibration curves prepared in methanol provide the same results that calibration curves prepared in Syrspend. As calibration curves prepared in methanol were obtained using geometric dilution, we wanted to do the same for syspend. Therefore we had to prepare an initial solution of syrspend and then perform the calibration curves by geometric dilution using methanol solution containing 10% of the suspending vehicle in order to mimic the exact composition of samples extract with methanol.

6. What is the signal to noise ratio at the lowest concentration or so called LOD?

According to reviewer’s suggestion, signal to noise ratio have been added in the revised manuscript (Table 2).

7. Lines 174/175 and 186. “Error!” message. Recheck.

This error has been corrected in the formatted manuscript

8. Degradation study: Supply chromatograms! What is the resolution of main peak from degradants? Provide a table with chromatogram attributes supporting your validation including RRTs for major degradants, resolution, tailing factor, theoretical plates, etc.

According to reviewer’s suggestions, chromatograms (Figures 1, 2, 3 and 4) have been added in the revised manuscript. Moreover, a table (Table 4) including parameters describing degradation byproducts has been provided.

Reviewer 4 Report

This is an interesting work evaluating the feasibility and stability of four APIs, which points a potential direction to liquid oral dosage medicine for neonates. I have several main questions:

1.       Is 60 days enough for the physical and chemical stability of the four API based oral liquid dosage forms?

2.       The introduction needs to be improved. The author mentioned that several methods to compound oral liquid preparation was reported in literatures. How are those methods comparing to the method in present manuscript? Why are the four APIs in present manuscript selected, but not other APIs? All these should be analyzed and introduced in introduction.

3.       The data obtained in section 2 should be presented as much as possible to make the discussion convincing. For example, from line 219 to line 222, the color modification, precipitates or suspendability, the PH change, osmolality change should be presented in terms of figure or tables of exact data, but not simply introduced by a sentences.

Author Response

Thank you very much for your valuable and constructive suggestions. We have carefully revised the manuscript according to the suggestions

1.       Is 60 days enough for the physical and chemical stability of the four API based oral liquid dosage forms?

Actually, the physicochemical stability of the four oral liquid dosage forms was investigated for a period of 60 days since the microbiological stability of the suspending agent has not been assessed over a longer period. This kind of period of stability is frequent for compounded oral formulations and in practice 60 days is suitable to the care of newborns.

2.       The introduction needs to be improved. The author mentioned that several methods to compound oral liquid preparation was reported in literatures. How are those methods comparing to the method in present manuscript? Why are the four APIs in present manuscript selected, but not other APIs? All these should be analyzed and introduced in introduction.

The introduction has been improved according to reviewer’s suggestions. We have introduced these points in the introduction (lines 52-55) and several parts of the manuscript discuss these points (lines 335-337, lines 342-343). The choice of the selected APIs was already discussed in the manuscript (lines 48-51).

3.       The data obtained in section 2 should be presented as much as possible to make the discussion convincing. For example, from line 219 to line 222, the color modification, precipitates or suspendability, the PH change, osmolality change should be presented in terms of figure or tables of exact data, but not simply introduced by a sentences.

According to reviewer’s suggestion, we have provided a figure (Figure 6) and a table (Table 5) in order to better describe physical parameters.

Round 2

Reviewer 3 Report

The authors have made changes to the manuscript as requested, although the novelty/originality aspect of this paper is still unclear. The formatting is highlighted in French and error messages still persist. English language should be rechecked (e.g. Table 5 title ends with “whatever storage”.. this is very informal). From an editorial standpoint, there are still major fixes to this manuscript. The authors should provide a reference to show that hypo osmolal dispersions are indeed safe as claimed by the authors.

Author Response

The authors have made changes to the manuscript as requested, although the novelty/originality aspect of this paper is still unclear.

We appreciate the comments of the reviewer. Many efforts have been made to be more compelling about this point:

Lines 350-352 « To our knowledge, the study presented in this paper is the first to assess the feasibility of compounding dexamethasone, phenytoin, hydrochlorothiazide and spironolactone oral suspensions using excipients free of potentially harmful effects »

Lines 362-366 « Once again, to our knowledge, the present study is the first to demonstrate the physicochemical stability of oral liquid dosage forms containing the targeted API and compounded without preservatives such as parabens. This result is of particular interest since the lack of preservatives may influence the chemical stability of a compound, especially due to their potential antioxidant activity »

In addition, potentiall harmful excipients, including endocrine disruptors, are, in our opinion, highly topical issues.

The formatting is highlighted in French and error messages still persist. English language should be rechecked (e.g. Table 5 title ends with “whatever storage”.. this is very informal). From an editorial standpoint, there are still major fixes to this manuscript.

We apologize for this point and we will deal it with the assistant editor. The initial manuscript was read and edited by Jefferey Arsham, American scientific translator, but another proofreading has been performed.

The authors should provide a reference to show that hypo osmolal dispersions are indeed safe as claimed by the authors.

No data has ever reported any troubles regarding oral ingestion of hypo osmolar solutions, in contrast to hyperosmolar ones. Moreover, in case of acute gastroenteritis oral rehydration with hypoosmolar solution is the major treatment and is recommended by the WHO (Guarino et al. J Pediatr Gastroenterol Nutr. 2014 Jul;59(1):132-52. doi: 10.1097/MPG.0000000000000375 ; WHO/UNICEF,  Joint  Statement:  Clinical  Manage-ment of Acute Diarrhoea - WHO/FCH/CAH/04.07- 2004,  World  Health  Organization). Finally, osmolality of water is low and oral ingestion of water has never been considered as unsafe, at least for small volumes.

Reviewer 4 Report

Significant improvement has been made to the manuscript. It can be accepted for publication now. 

Author Response

We sincerely appreciate the comments of the reviewer.